# Using a Very Low Energy Diet to Achieve Substantial Preconception Weight Loss in Women with Obesity: A Review of the Safety and Efficacy

**DOI:** 10.3390/nu14204423

**Published:** 2022-10-21

**Authors:** Sarah A Price, Priya Sumithran

**Affiliations:** 1Department of Obstetric Medicine, Royal Women’s Hospital, Flemington Rd, North Melbourne, VIC 3051, Australia; 2Department of Medicine, University of Melbourne, Grattan St, Parkville, VIC 3010, Australia; 3Department of Diabetes and Endocrinology, Royal Melbourne Hospital, Grattan St, Parkville, VIC 3010, Australia; 4Department of Endocrinology, Austin Health, Waterdale Rd, Heidelberg Heights, VIC 3081, Australia

**Keywords:** obesity, preconception, pregnancy, very low energy diet (VLED), weight loss, ketosis

## Abstract

Obesity in women of reproductive age is common. Emerging evidence suggests that maternal obesity not only increases the risk of adverse pregnancy outcomes but also has an enduring impact on the metabolic health of the offspring. Given this, management of obesity prior to pregnancy is critically important. Almost all international guidelines suggest that women with obesity should aim to achieve weight loss prior to pregnancy. However, current pre-conception weight loss therapies are sub-optimal. Lifestyle modification typically results in modest weight loss. This may assist fertility but does not alter pregnancy outcomes. Bariatric surgery results in substantial weight loss, which improves pregnancy outcomes for the mother but may be harmful to the offspring. Alternative approaches to the management of obesity in women planning pregnancy are needed. Very low energy diets (VLEDs) have been proposed as a possible tool to assist women with obesity achieve weight loss prior to conception. While VLEDs can induce substantial and rapid weight loss, there are concerns about the impact of rapid weight loss on maternal nutrition prior to pregnancy and about inadvertent exposure of the early fetus to ketosis. The purpose of this review is to examine the existing literature regarding the safety and efficacy of a preconception VLED program as a tool to achieve substantial weight loss in women with obesity.

## 1. Introduction

Obesity in women of child-bearing age is common. It is predicted that over the next decade, the greatest increase in the prevalence of obesity will occur in women of reproductive age [1]. This is important, given the strong and continuous association between maternal body mass index (BMI) and adverse pregnancy outcomes [2,3]. Furthermore, it is widely appreciated that the early metabolic environment of the offspring influences its susceptibility to metabolic disease later in life [4,5,6,7].

The short-term risks of maternal obesity are well documented. These include miscarriage [8], gestational diabetes [9], gestational hypertension and pre-eclampsia [9,10], Caesarean section [11], anaesthetic complications and wound infections [2]. Mothers with obesity are less likely to initiate and maintain breastfeeding [12]. Neonatal risks of maternal obesity include large-for-gestational age, hyperbilirubinemia and hypoglycaemia [2]. Maternal obesity confers an elevated risk of congenital anomalies [13], especially congenital heart disease and neural tube defects [14]. The risks of preterm delivery [15] and of neonatal death are also increased [16].

Obesity in pregnancy also has long-term impacts for both mother and child. For the mother, obesity during pregnancy is an independent risk factor for later cardiovascular disease. Lee et al. found that after a median follow-up of 73 years, women with obesity during pregnancy (BMI > 30 kg/m^2^) had an increased rate of major cardiovascular events (HR 1.26; 95% CI 1.01–1.57) and all-cause mortality (HR 1.35; 95% CI, 1.02–1.77) compared to women with a normal BMI during pregnancy even after adjustment for confounding factors [17].

For the offspring, the first 1000 days of life from conception to around 2 years of age, is a critical period of developmental plasticity [18]. Exposure to excessive nutrients impacts the growth, body composition and metabolic profile of the offspring [7,19]. This has been termed ‘fuel-mediated teratogenicity’ [20]. In addition to the short-term effects on the fetus, the maternal metabolic milieu likely also causes epigenetic changes in the offspring resulting in a predisposition to metabolic disease later in life [21].

The three most significant risk factors for obesity in childhood are maternal obesity [22], maternal gestational diabetes [5] and being born large or small for gestational age (LGA or SGA). Of children with these risk factors, half will develop features of the metabolic syndrome by 6 years of age [23,24], perpetuating the cycle of obesity [23,24]. Given that antenatal interventions have not been successful in addressing this critical issue of metabolic programming [25,26], there is increasing emphasis on pre-pregnancy interventions [27,28].

Guidelines from several countries recommend women with obesity should institute pre-conception lifestyle change, comprising an energy-reduced diet and increased physical activity [29]. This typically results in modest weight loss (<3% body weight) which may improve fertility but does not impact pregnancy outcomes [30,31]. Bariatric surgery achieves substantial weight loss (>15% body weight) which reduces the rate of maternal adverse pregnancy outcomes such as gestational diabetes and gestational hypertension [32] but may increase the rate of neonatal adverse outcomes such as small for gestational age and possibly neonatal mortality [16,32].

Very low energy diets (VLEDs) are broadly defined as diets which provide less than 3.4 MJ (800 kcal) per day, and which contain daily allowances of protein and essential micronutrients. If used as the only source of nutrition, or in combination with very low carbohydrate intake, they result in ketosis [33]. These are distinct from ketogenic diets which are high in energy due to a high content of fat and protein.

VLEDs have been used in clinical practice for more than 20 years. If used as prescribed, a weight loss of 1.5–2.5 kg per week, and 10–15% body weight loss within 12 weeks, is expected [34]. VLEDs are a potential tool for the management of obesity in women planning pregnancy. Although they have numerous characteristics that make them particularly suitable for weight loss in the pre-pregnancy period, including the potential to induce rapid and substantial weight loss, there is also concern that the resulting rapid weight loss could have deleterious consequences on the offspring [35]. Further, there is concern about inadvertent pregnancy while using a VLED which may expose the early fetus to ketosis [36].

The purpose of this review is to examine the existing literature regarding the suitability of a VLED as a tool to achieve weight loss in women with obesity planning pregnancy.

## 2. Basic Principles of the Use of Ketogenic Diets for Weight Loss

The term ‘ketones’ refers to three molecules: beta-hydroxybutyrate (BHB), acetoacetate (AcAc) and acetone [37]. The circulating level of ketones depends on the rate of production (ketogenesis) and the rate of utilisation (ketolysis). Ketogenesis occurs mostly during fasting or prolonged starvation and is suppressed in the post-prandial state. Most ketone bodies are generated in the mitochondria of hepatocytes after the break-down of fatty acids derived from adipose tissue [37]. This is stimulated primarily by beta-adrenergic catecholamines and glucagon, and the process is inhibited by insulin [37]. Ketolysis is the conversion of ketones to acetyl CoA for energy production. This process occurs in the mitochondria of extra-hepatic organs.

The body stores sufficient carbohydrate as glycogen in the muscle and liver to provide energy for approximately three days [38,39]. Over three days of either starvation or omission of carbohydrate from the diet, blood and urinary ketones progressively rise. If this dietary pattern is maintained for 5–6 weeks, blood ketones plateau around 8 mmol/L [39]. The plateau may occur because steady state is reached between increased ketone body utilisation by the brain, and decreased ketone body utilisation by skeletal muscle which subsequently uses free fatty acids as the principal source of fuel [39].

### 2.1. Definition of Ketogenic Diet

There is no precise definition of a ketogenic ‘very low energy diet (VLED)’. Most sources would suggest these diets contain around 800 calories per day which usually includes less than 50 g of carbohydrate and a moderate intake of protein [40,41,42]. However, the carbohydrate intake required to induce ketosis is vastly different between individuals [43]. Some studies suggest ketosis can occur even when greater than 150 g carbohydrate per day is consumed [44,45,46]. Conversely, other studies report that even when a very low carbohydrate diet is consumed (<30 g/day) less than half of participants developed ketones in urine [47]. The presence of urinary ketones correlates well with the presence of blood ketones during a VLED [48]. However, they do not correlate well with the amount of weight loss [48].

### 2.2. Importance of Nutritional Composition of Ketogenic Diets

Multiple studies have demonstrated that the restriction of dietary carbohydrate, and not restriction of calories, is responsible for initiating the metabolic response to short-term fasting including ketosis [49,50,51]. Carbohydrate restriction forces a shift from glucose to fatty acid metabolism resulting in ketosis [52]. Conversely, ketosis can be eliminated by administration of carbohydrate even when calorie requirements remain unmet [53].

The protein content of the diet is important in determining if ketosis will be achieved. Diets that are very high in protein can limit ketosis because the protein acts as a gluconeogenic substrate [54]. When eating a low carbohydrate high protein diet, up to 50 g of glucose can be derived from 100 g of protein. This is sufficient to limit ketosis [55]. Conversely, low protein diets intended to induce weight loss may result in excessive loss of fat free mass [56]. A VLED program aims to provide at least 0.8 g protein per kilogram lean body weight per day. This is sufficient to prevent excessive loss of fat free mass during weight loss but is not sufficient to prevent the development of ketosis [41,57].

### 2.3. Ketosis over the Course of a Ketogenic Dietary Program

Sumithran et al. showed that the rise in ghrelin after weight loss did not occur while weight-reduced participants were ketotic. Subjective appetite ratings were also lower during ketosis [38,58]. This data, and that of other studies including administration of ketones [43,59], suggest that ketosis may facilitate dietary adherence by influencing hormones which control appetite.

Over the course of a VLED program, Nymo et al. report that the impact of ketosis on appetite suppression is not stable. Over a 13-week VLED program resulting in ketosis, they report a significant increase in fasting hunger was observed by day 3 with intermittently increased drive to eat up to three weeks (5% weight loss). After that, and while participants are ketotic, a 10–17% weight loss was not associated with increased appetite [60].

### 2.4. Ketosis in the Pre-Pregnancy Period

Ketone body metabolism in women planning pregnancy is no different to the general adult population [57]. If a ketogenic diet is ceased and carbohydrates reintroduced, ketones are cleared within hours [38]. However, studies suggest that around half of pregnancies are unplanned [61,62]. Even amongst women involved in pre-conception health activities, inadvertent pregnancies inevitably occur [63]. Furthermore, in women with obesity, recognition of pregnancy may be delayed due to irregular menses and body habitus [64]. Although this review focuses on the suitability of a VLED program as a tool for weight loss in women with obesity planning pregnancy, inadvertent conception means the impact on the early fetus must also be considered.

### 2.5. Ketosis in Pregnancy

The tendency towards ketosis is never greater during adult life than in pregnancy [65,66]. Compared with a non-pregnant woman where ketones are produced after 24–36 h of fasting, ketones can occur after 16 h of fasting in a woman in both mid and late pregnancy [67,68], because the feto-placental unit acts as a drain on maternal glucose. The subsequent fall in glucose suppresses insulin secretion which enhances lipolysis and the conversion of fatty acids to ketones [67]. Further, the increased insulin resistance of pregnancy decreases the ability of insulin to suppress lipolysis [67].

Ketones readily cross the placenta in humans (Figure 1). This information is known because of two observations. Firstly, when women in early pregnancy were fasted for up to 84 h prior to elective termination of pregnancy, beta-hydroxybutyrate and acetoacetic acid levels in amniotic fluid rose by up to 10–30 fold which was proportional to levels in maternal blood [67]. Secondly, ketones are elevated in amniotic fluid of women with poorly controlled diabetes [67]. The placenta contains enzymes necessary for the conversion of acetyl co-enzyme A to hydroxybutyrate and hence the placenta might contribute further to the supply of ketones reaching the fetus. The placenta also modulates the passage of other nutrients to the fetus [67,69].

Although the fetus may be exposed to ketone bodies from the maternal circulation, the fetus is rarely exposed to severe ketosis even during maternal starvation. The fetus can adapt to utilise ketone bodies as an energy source for the brain, liver and kidneys [67]. In addition to use as an energy source, some studies suggest that ketone bodies can be used as a substrate in lipid and myelin synthesis. In animal studies, ketone bodies injected subcutaneously into 7–12 day old rats are subsequently incorporated into the myelin in the brains of these rats [67]. This evidence suggests that mild, transient ketosis is physiological and is unlikely to be deleterious to the developing fetus

## 3. Rationale for Suitability of a VLED Diet in the Pre-Pregnancy Setting

In women with obesity planning pregnancy, the ideal intervention would be relatively short in duration, efficacious in achieving sufficient weight loss to improve health prior to conception, and would not compromise maternal nutrition. For this reason, VLEDs have been proposed as potential tool for pre-pregnancy weight loss.

### 3.1. Rationale for Suitability

In a general population of adults with obesity, a VLED program will result in 10–15% body weight loss within 12 weeks [70]. To establish the efficacy of VLEDs in the pre-pregnancy period, we conducted a review of the literature. We aimed to include randomised controlled trials and non-randomised studies. Only studies published in English after January 2000 were included. A search was performed of the following electronic databases from January 2000 to January 2022: MEDLINE (Ovid), Embase (Ovid), CINAHL (EbscoHost), Scopus and Web of Science. The search criteria are shown in Appendix A. To screen for study inclusion by one author, COVIDENCE software was used. Of the 112 studies identified, 101 were excluded after abstract review and a further 4 studies were excluded after full-text review. The PRISMA flowchart is shown in Figure 2.

Seven published studies met the inclusion criteria of using a VLED program to achieve weight loss prior to pregnancy in women with obesity (Table 1) [71,72,73,74,75,76,77,78] with one study being published in two parts [74,75]. Four of the included studies were cohorts of women planning pregnancy using in vitro fertilization (IVF), three were pilot studies [76,77] and one was a randomised controlled trial in women planning spontaneous conception.

Reported pre-pregnancy weight loss was highly variable (3.8–17.1 kg), likely reflecting differences in study protocol and the duration of the VLED program. Although the studies that aimed to achieve ketosis in the VLED participants can be identified (as shown in Table 1), it is not possible to determine the carbohydrate intake in each of the studies nor whether individual participants achieved ketosis during the study. Despite these limitations, the available data would suggest VLEDs are an efficacious weight loss tool in women with obesity planning pregnancy.

### 3.2. Tolerability and Acceptability of VLEDs in Women Planning Pregnancy

Previous reports suggest that VLEDs cause only transient mild side effects such as halitosis, headache, cold intolerance, constipation and postural dizziness [40]. Ketogenic VLEDs also assist to suppress hunger during weight loss [57].

For women of advanced maternal age or where other co-morbidities may reduce fecundity, a weight loss program will only be practical if it is of short duration, allowing adequate time to achieve conception [79]. VLEDs fulfil this criterion of acceptability. Although not well studied, there is some evidence that rapid weight loss over a short period of time is a motivating factor in continued weight loss [40].

### 3.3. VLEDs Support Maternal Nutrition

VLEDs have been designed to contain adequate protein and micronutrients to meet nutritional requirements during weight loss [80]. They also should contain sufficient micronutrients and fatty acids including omega-3 (alpha-linoleic acid) and omega-6 (linolenic acid) to meet the 2020 World Health Organisation recommendations for women planning pregnancy [40,81,82]. Usual preconception folate supplementation is still recommended, as is supplementation for any specific nutritional deficiencies [81].

VLEDs should contain adequate protein to maintain muscle mass during weight loss. In an elderly population, Haywood et al. demonstrated that a VLED resulting in ketosis protected lean muscle during a weight loss program better than a standard dietary intervention [56]. Some studies suggest that ketones have anti-catabolic effects [83,84] although this is uncertain.

When pregnancy is desired, the VLED program should be ceased. A period of weight stabilisation is suggested prior to attempting to conceive. This allows women to clear ketones and recommence a regular food diet in accordance with dietary guidelines before conception [85]. Given that VLEDs do not alter gastrointestinal anatomy, as would occur with surgical management of obesity, no restriction of gestational weight gain is expected.

### 3.4. VLEDs Engage Patients in Pre-Pregnancy Weight Loss

The lack of evidence-based weight loss tools for the pre-conception period may be a deterrent to women with obesity engaging with health care professionals when planning pregnancy. It may also be a barrier to health care professionals addressing weight management when women present for pre-conception care [63]. If safe, effective and well-tolerated tools for achieving weight loss prior to conception are available, this may assist women to engage with health care professionals both for weight management and for other aspects of pre-conception care.

## 4. Rationale for Concerns about a VLED in the Pre-Pregnancy Setting

Animal studies and observational data have raised concerns about inadvertent exposure of the early fetus to ketosis in the setting of unplanned pregnancy, and the weight trajectory of women after cessation of the VLED.

### 4.1. Neurological Effects of Ketosis on the Fetus

Although transient ketosis in pregnancy is widely accepted as physiological (i.e., normal pregnancy, maternal hyperemesis, starvation, diabetes) there are concerns about the use of ketogenic diets prior to pregnancy, where an inadvertent pregnancy may result in the very early fetus being exposed to ketosis.

Early studies in the 1960’s suggested that exposure to ketosis during pregnancy was associated with a reduced IQ in the offspring [86]. However, these studies are now recognised as flawed due to the irregular intervals of testing ketosis, inconsistent protocols for urine and/or blood ketone testing, the retrospective nature of the studies and failure to control for confounding factors such as uncontrolled diabetes and nutritional status of the mother. Indeed when the data for the study by Churchill et al. were re-analysed, taking into account confounding factors, no association between ketosis and neurological status of the offspring was found [67].

In 1991, Rizzo et al. reported that after correction for socioeconomic status, race and ethnic origin, children born to women with pre-existing and gestational diabetes had Bayley Scale scores (mental development index scores at age 2 years) and Stanford-Binet scores (a measure of intelligence at age 3–5 years) which correlated inversely with beta-hydroxybutyrate concentrations in the third trimester of pregnancy [87]. Although these studies were performed in women with diabetes during pregnancy compared with controls, the potential risk of ketosis on neurological development cannot be completely discounted.

In animal studies, ketosis is associated with reduced myelin in the brains of offspring. Sussman et al. explored this concept by feeding female mice a ketogenic diet for the entire gestation of pregnancy and during lactation. MRI of the brains of offspring showed a relative bilateral decrease in the cortex, fimbria, hippocampus, corpus callosum and lateral ventricle, but a relative volumetric enlargement of the hypothalamus and medulla [36]. This is thought to be due to changes in myelin synthesis in the setting of ketosis. Importantly, these changes occurred in the context of severe ketosis (up to 4 mmol/L) for the entire period of gestation and lactation [36]. These studies are at odds with earlier studies suggesting ketone bodies may be used as an energy source to produce myelin. However, these differences may be explained by differences in the severity and duration of ketosis.

The Sussman et al. studies [36,88] do not reflect the transient exposure to ketosis that could inadvertently occur using a VLED program prior to pregnancy. Human models that more closely represent this scenario are hyperemesis gravidarum [89] and type 1 diabetes. In both cases, transient mild ketosis is not associated with adverse neurological outcomes in offspring [90].

### 4.2. Metabolic Effects of VLED on the Fetus

As noted by Matasiuk et al., epidemiological and animal studies suggest that offspring conceived during a time of maternal nutritional restriction may be ‘programmed’ for metabolic disease [35]. This concept is referred to as the ‘thrifty phenotype’ and may include changes in the hypothalamic-pituitary-adrenal axis, body composition, glucose metabolism, and cardiovascular function [35].

Although both starvation and VLED programs result in ketosis, these states are fundamentally different. During starvation, inadequate intake of protein results in protein malnutrition. Protein malnutrition has been shown to result in both adverse pregnancy outcomes and adverse metabolic programming [81].

Grieger et al. have demonstrated that protein malnutrition during pregnancy is associated with an increased risk of preterm birth [91]. Using an animal model, Josse et al. demonstrated that 7-month old offspring of mothers who ate a low (10%) protein diet during pregnancy had a lower muscle mass and higher food intake than mice born to mothers who consumed a control (22% protein) diet [92]. Similar findings were reported in the study by Qusam and colleagues [93]. Rocha et al. suggest that protein malnutrition during pre-conceptual life alters neuropeptide Y receptor distribution along the arcuate/paraventricular pathway, which may be associated wtih altered appetite regulation in later life [94,95]. A VLED program meets nutritional requirements for both protein and micronutrients [40,41] and hence this adverse programming is not expected.

### 4.3. Growth Effects of VLEDs on the Fetus

One of the primary concerns around pre-conception weight loss is the possibility of increasing the rate of offspring that are small for gestational age (SGA). SGA is often used as a proxy for identification of intra-uterine growth restriction (IUGR) [35]. These infants are at increased risk of both short- and long-term morbidity. Emerging evidence has shown IUGR offspring are likely to have lower endowment of cardiomyocytes [96], nephrons [97] and pancreatic beta cells [98], which increases the long-term risk of metabolic disease. Furthermore, there is strong evidence that being born IUGR predisposes to long-term insulin resistance and adiposity [99,100,101,102].

Some retrospective studies of pregnancy outcomes in women after bariatric surgery indicate a higher rate of SGA infants [30,32]. It is commonly presumed that this is the result of caloric restriction [103]. However, the rate of SGA offspring is higher when the interval between surgery and conception is 12–24 months rather than during the initial phase of rapid weight loss (<12 months) [32,104]. This suggests that it is not the caloric restriction per se that results in an increased rate of SGA but rather may be related to sub-optimal maternal nutrient absorption during pregnancy [32,105]. In contrast, non-surgical pre-conception weight loss programs such as a VLED do not alter the gastrointestinal anatomy. Therefore, during pregnancy, normal nutrient absorption is anticipated

### 4.4. Weight Maintenance after a VLED Program

It is important that weight loss achieved during a pre-pregnancy VLED program is maintained until conception, and is not associated with subsequent excessive gestational weight gain.

Purcell et al. demonstrated that the rate of weight loss does not impact on the rate of weight regain [106]. Therefore rapid weight loss after a VLED program is not expected to result in more rapid weight regain than weight loss achieved with lifestyle modification. In studies of adults with obesity who were not planning pregnancy, less than half of the weight lost achieved with a VLED was regained 12-months after cessation of the intervention [58,70].

In the only study to report weight maintenance after a preconception VLED program, there was no difference between weight regain prior to coneption in women who achieved weight loss through lifestyle modification compared with a VLED (3.0 vs. 3.6 kg, *p* = 0.84) and also no difference in gestational weight gain (10.9 vs. 10.3 kg, *p* = 0.66).

## 5. Recommendations and Further Research

There are insufficient studies to determine the role of VLEDs in the pre-pregnancy period. However, the available literature does not suggest evidence of harm. Concern around the use of VLEDs prior to pregnancy has likely occurred because data has been extrapolated from studies that do not reflect the use of a VLED program prior to pregnancy.

In order to clarify the impact of a VLED prior to pregnancy, large randomised controlled studies would be ideal. These studies could have two purposes. Firstly, they could assess if this tool is acceptable, efficacious and safe for both mother and offspring. Secondly, they could determine if the weight loss induced by using a VLED program prior to pregnancy reduces the rate of obesity-related adverse pregnancy outcomes.

Given that large stand-alone randomized trials are challenging, slow and costly, particularly when aiming to recruit women in the pre-conception period, alternative approaches may be needed. Establishing pragmatic clinical trials that are embedded in clinical practice or data linkage studies may assist in building the evidence to better understand the place of VLEDs in the management of obesity prior to pregnancy.

## 6. Conclusions

Very low energy diets may be a suitable tool for use in the pre-pregnancy setting. Concerns about the use of VLEDs prior to conception have been raised due to the potential impact on the neurological status, growth and metabolic status of the offspring. However, current evidence does not support these concerns. Similarly, concerns about the adverse weight trajectory of women after cessation of a VLED are not supported by the current evidence.

VLEDs are an efficacious and well tolerated tool that may be used to achieve pre-conception weight loss in women with obesity. VLEDs support maternal nutrition and may assist to engage women with obesity in pre-conception interventions. Further evaluation of the efficacy, tolerability and safety of VLEDs in the pre-pregnancy setting is required to gather sufficient evidence to guide clinical care in the pre-conception period.

## Figures and Tables

**Figure 1 nutrients-14-04423-f001:**
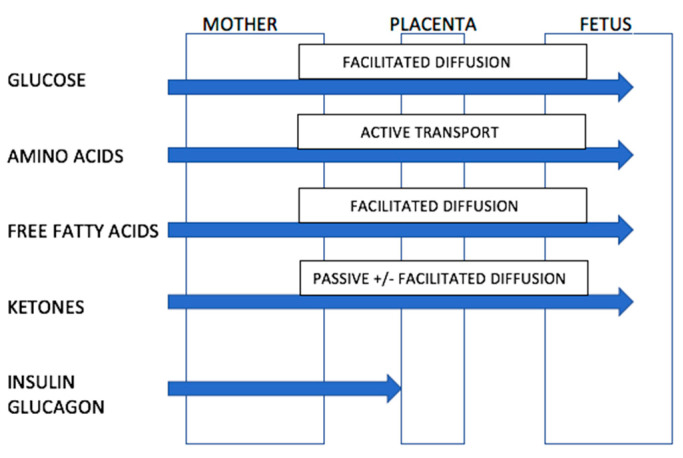
Ability of nutrients and ketones to transverse the placenta. Source: Adapted from Rudolf, M.C.; Sherwin, R.S. [67]. (Nb. Transport of free fatty acids across the placenta remains controversial but may occur via facilitated diffusion. The placenta may protect the fetus from excessive fatty acid exposure through esterification of free fatty acids to triglycerides).

**Figure 2 nutrients-14-04423-f002:**
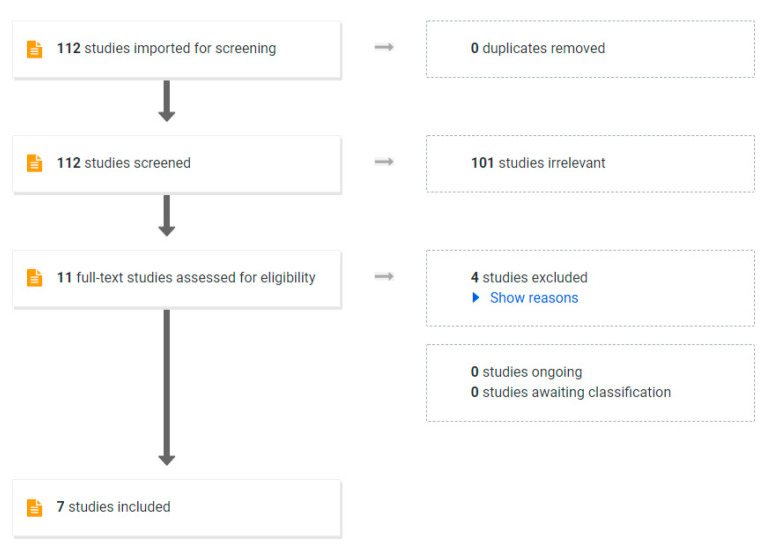
PRISMA Flow chart used in systematic review.

**Table 1 nutrients-14-04423-t001:** Summary of studies that used a VLED program as a tool to achieve weight loss in women with obesity prior to pregnancy.

Author, Year.	Study Design	Sample Size (Case/Control)	Cohort	Designed to Achieve Ketosis	Duration VLED (Wks)	Duration wt Maintenance (Wks)	Weight Loss (kg) (Case/Control)	Pregnancy Rate (%) (Case/Control)	Live Birth Rate (%) (Case/Control)	Pregnancy Outcomes Reported (Y/N)
Tsagareli et al., 2006 [71].	Pilot	N = 10	IVF	Y	3–6	0	5.6	0	0	N
Moran et al., 2011 [72].	RCT	N = 46 (21/25)	IVF	N	7	0	3.8 ± 3.0/ 0.5 ± 1.2	67/40	39/25	N
Sim et al., 2014 [73].	RCT	N = 49 (27/22)	IVF	Y	6	6	6.6 ± 4.6/ 1.6 ± 3.6	48/14	44/14	N
Einarsson et al., 2017 & 2018 [74,75].	RCT	N = 317 (160/157)	IVF	Y	16	2–5	9.44 ± 6.57/ 1.19 ± 1.95	Not specified	45/41	Y
Rothberg et al., 2016 [76].	Pilot	N = 14	Prepreg	Y	12	4	14 ± 6/ 5 ± 5	3/0	3/0	N
Brackenridge et al., 2018 [77].	Pilot	N = 14 (6/8)	Prepreg	N	26	0	17.1/2.7	Not specified	Not specified	N
Price et al., 2021 [78].	RCT	N = 164 (85/79)	Prepreg	Y	12	4	13.0/3.2	67/46	43/28	Y

## Data Availability

Not applicable.

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
