# Peer review of "Using a Very Low Energy Diet to Achieve Substantial Preconception Weight Loss in Women with Obesity: A Review of the Safety and Efficacy"

_nutrients, 2022, doi:10.3390/nu14204423_

Round 1

Reviewer 1 Report

Using a Very Low Energy Diet to achieve substantial preconception weight loss in women with obesity: A review of the safety and efficacy

Dear Authors,

Congratulations on writing such an interesting article to examine the current literature regarding the safety and efficacy of very low energy diet (VLED) programs in the preconception period.

The following are my comments and suggestions:

 1. Kindly provide the information about the VLED compositions and fatty acids quantity in the VLED, as recommended by WHO in brief.

2. Does any observation notice in fetuses if they get exposed to ketone bodies? Please describe it in detail in respect of the molecular pathways.

3. I was wondering why the author did not present or show the withdrawal effect of VLEDs. Kindly provide the information about it and other cytokines modulated during the switch of diets.

4. table 1 - Kindly provide the percentage of carbohydrates used in the respective diets/studies, which was mentioned in table 1.

5. 4.1.1-So, the fetus was developing normally without any health consequences when get exposed to ketone bodies? Any health issues were mentioned?

7. How does the degree of effects determined when a fetus is exposed to ketones? by the scoring system? Can the author present the data for this scoring?

8. References – references are not consistence. The journal name is not the same throughout the document. Some are either in short form or in long form. Kindly fix it.

Discussion:

Kindly provide brief details about the metabolism of ketone bodies and how they get modulated or affects in case of obesity conditions. Kindly explain the secondary metabolites originated from or via ketone.  

What is the most interesting feature that makes the VLED or keto diet a potential therapeutic agent? Kindly use a few bioinformatics tools and please predict the GO and KEGG pathways that get affected by the Keto diet.

Author Response

Please see the attached word document.

Reviewer 2 Report

It is unclear if this document was a part of a larger document.  Many references are not used in the body of the text, and many of the articles used as part of the literature review were not even discussed within the manuscript. Please consider this along with my comments. 

Using a Very Low Energy Diet to achieve substantial preconception weight loss in women with obesity: A review of the safety and efficacy

Dear author,

Thank you for the opportunity to review this article.  This is an essential and timely research topic.  My comments and feedback are included for consideration.

Introduction:

-Section 2. Rationale for suitability of a VLED diet in the pre-pregnancy setting.

Current line reads “When carbohydrate intake is restricted, the body is forced to used stored fat as the primary energy source, resulting in a rise in the circulating level of ketones”(15)

Need to change “used” to “use”

-The very next line…”Sumithran and colleagues…” and reference is listed as 15.  However, on the reference list Sumithran is number 33.  Please double check references and their order.

-Additionally, Haywood is listed as 18 in the text but then on the references list it is 43.

-Regarding the table: Likely the reference number vs. year of citation should be listed in the table as well.

-Sections 3.1 a d 4.1 seems to have some erroneous punctuation prior to the header

-Section 3.1.2 . Metabolic effects of VLEDs on the fetus There are concerns that maternal weight loss prior to pregnancy may result in adverse metabolic programming in the offspring akin that that seen after.

Remove the second “that”

Further in that same section “…composition (60) (61) (62)”…. Should read “…composition (60-62).”

Results or Findings:

Also following table, each study should be discussed in detail.  Only finding a reference to and discussion of the Price and Einarsson articles and their findings. Additionally, much of the content following the table has nothing to do with the tables results (i.e. Sussman etc.). This may also be partly due to the refences being so unorganized and hard to follow.

Conclusion:

This conclusion should be far more robust than these few sentences.

Table 1

References in Table 1 do not appear correct (80, 81, 82 etc.)

Figures:

There is a reference to a Prisma diagram, but one is not included. Figure S1.

References:

Approximately references listed as 34-59 do not appear to be mentioned in the manuscript, therefore should be removed from the reference list.

Additionally, references 72-88 do not appear in the manuscript and should be removed from the reference list.

Round 2

Reviewer 2 Report

The concerns I had with this manuscript have been adequately addressed.